# Runx3 Restoration Regresses K-Ras-Activated Mouse Lung Cancers and Inhibits Recurrence

**DOI:** 10.3390/cells12202438

**Published:** 2023-10-11

**Authors:** Ja-Yeol Lee, Jung-Won Lee, Tae-Geun Park, Sang-Hyun Han, Seo-Yeong Yoo, Kyoung-Mi Jung, Da-Mi Kim, Ok-Jun Lee, Dohun Kim, Xin-Zi Chi, Eung-Gook Kim, You-Soub Lee, Suk-Chul Bae

**Affiliations:** 1Department of Biochemistry, School of Medicine, Institute for Tumor Research, Chungbuk National University, Cheongju 28644, Republic of Korea; izzal2000@chungbuk.ac.kr (J.-Y.L.); jeongwon@chungbuk.ac.kr (J.-W.L.); tgeun@chungbuk.ac.kr (T.-G.P.); hansh@chungbuk.ac.kr (S.-H.H.); ysy9533@chungbuk.ac.kr (S.-Y.Y.); jungkm24@chungbuk.ac.kr (K.-M.J.); damikim@kaeri.re.kr (D.-M.K.); xinzi@chungbuk.ac.kr (X.-Z.C.); egkim@chungbuk.ac.kr (E.-G.K.); 2Department of Pathology, School of Medicine, Chungbuk National University and Hospital, Cheongju 28644, Republic of Korea; ojlee@chungbuk.ac.kr; 3Department of Thoracic and Cardiovascular Surgery, School of Medicine, Chungbuk National University and Hospital, Cheongju 28644, Republic of Korea; mwille@chungbuk.ac.kr

**Keywords:** runt-related transcription factor 3 (Runx3), K-Ras, p53, lung cancer, resistance, recurrence

## Abstract

Oncogenic *K-RAS* mutations occur in approximately 25% of human lung cancers and are most frequently found in codon 12 (G12C, G12V, and G12D). Mutated K-RAS inhibitors have shown beneficial results in many patients; however, the inhibitors specifically target K-RAS*^G12C^* and acquired resistance is a common occurrence. Therefore, new treatments targeting all kinds of oncogenic *K-RAS* mutations with a durable response are needed. RUNX3 acts as a pioneer factor of the restriction (R)-point, which is critical for the life and death of cells. *RUNX3* is inactivated in most *K-RAS*-activated mouse and human lung cancers. Deletion of mouse lung *Runx3* induces adenomas (ADs) and facilitates the development of *K-Ras*-activated adenocarcinomas (ADCs). In this study, conditional restoration of *Runx3* in an established *K-Ras*-activated mouse lung cancer model regressed both ADs and ADCs and suppressed cancer recurrence, markedly increasing mouse survival. *Runx3* restoration suppressed *K-Ras*-activated lung cancer mainly through Arf-p53 pathway-mediated apoptosis and partly through p53-independent inhibition of proliferation. This study provides in vivo evidence supporting *RUNX3* as a therapeutic tool for the treatment of *K*-*RAS*-activated lung cancers with a durable response.

## 1. Introduction

Lung adenocarcinoma (ADC) is the most frequent subtype of lung cancer. Most lung ADCs develop through stepwise progression from adenoma (AD) to ADCs [1,2]. Approximately 25% of human lung ADC cases harbor activating mutations in the *K-RAS* gene [3]. The most frequent mutations occur in codon 12, and the most common subtypes are G12C, G12V, and G12D. Recently, new drugs targeting a specific type of *K-RAS* mutation (*K-RAS^G12C^*) were conditionally approved by the USA Food and Drug Administration [4]. The inhibitors effectively regress *K-RAS^G12C^*-mutated lung cancers. However, the cancers commonly recur within 1 year. The acquired drug resistance is mainly due to secondary oncogenic mutations occurring at oncogenic *K-RAS* itself or at both upstream (*EGFR*, *HER2*, *FGFR*) and downstream (*MAPK*/*MEK* pathway) sites [5,6,7]. This rapid recurrence after inhibition of oncogenic K-RAS was observed previously in a mouse lung cancer model in which tumors initially responded to knockdown of oncogenic *K-Ras* but recurred after 2 weeks with secondary oncogene activation [8]. These results indicate that existing strategies for inhibiting oncogenic K-RAS have a limited ability to achieve a durable response in the context of cancer treatment.

To establish new treatments with a durable response, it is important to determine whether cells have evolved effective defense mechanisms against oncogene activation. Normal cells have a defense mechanism against oncogenic *K-Ras* involving the Arf-p53 pathway [9,10,11]. Simultaneous activation of *K-Ras* and inactivation of *p53* in the mouse lung accelerates malignant progression to ADC [12]. Considering the pro-apoptotic function of p53, restoration of *p53* is considered an attractive therapeutic intervention. However, *p53* restoration in *K-Ras*-activated mouse lung cancer suppresses ADCs but does not affect ADs, which are likely to develop into ADCs [13,14,15]. Consistent with this, heterozygous oncogenic *K-Ras* mutations induce lung AD/ADC in the absence of *p53* mutation [16], and loss of *p53* does not have a significant impact on early *K-Ras*-induced lung tumorigenesis [17]. Whether cells have evolved effective defense mechanisms against heterozygous oncogenic *K-Ras* mutations remains unclear.

A tumor is defined as an abnormal mass of tissue that forms when cells divide at a higher rate than normal or do not die when they should. The cellular decision regarding whether to undergo division or death is made at the restriction (R)-point, which is disrupted in nearly all tumors [18]. RUNX3 functions as a pioneer factor of the R-point and leads to a death decision in response to aberrant persistence of RAS signals [19,20,21,22,23]. When a death decision is made at the R-point, the cellular defense against tumorigenesis involves activating the ARF-p53 pathway [20]. *RUNX3* is deactivated by epigenetic alterations in most cases of *K-RAS*-activated mouse and human lung ADCs [24,25]. *Runx3* inactivation not only abrogates R-point-associated Arf-p53 pathway activity but also promotes the formation of lung ADs and accelerates the development of oncogenic *K-Ras*-dependent ADCs [25].

Conceivably, lung AD cells that develop as a result of *Runx3* inactivation are unable to defend against oncogenic *K-Ras*, resulting in the transition from AD to ADC upon oncogenic *K-Ras* mutation. However, it remains unclear whether *Runx3* can effectively regress already-established *K-Ras*-activated lung AD and ADC in vivo. To address this question, we developed a mouse model in which *Runx3* is conditionally restored by inducible Flippase. This mouse model was used to demonstrate that *Runx3* restoration effectively regresses established lung cancers and inhibits recurrence. The results indicate that cells have evolved an effective defense mechanism against heterozygous oncogenic *K-Ras* mutation. The mechanism is abrogated by *Runx3* inactivation and can be re-established with *Runx3* restoration. These results support the potential of *RUNX3* as a therapeutic tool for the treatment of *K-RAS*-activated lung cancers with a durable response.

## 2. Materials and Methods

### 2.1. Mice

*Runx3^flox^* (Jax 008773), *p53^flox^* (Jax 008462), *K-Ras^LSL-G12D^* (Jax 008179), *K-Ras^LA1^* (Johnson et al., 2001), *Rosa26R-Tomato* (Jax 007914), and *Flp^ERT2^* (*R26^FlpoER^*, Jax 019016) mice were obtained from Jackson Laboratory (Bar Harbor, ME, USA). *Runx3^Frt-Stop-Frt/+^* (*Runx3^FSF/+^*) mice originated from Macrogen (Seoul, Republic of Korea). All mice analyzed had mixed genetic backgrounds and were age matched (6–8 weeks old) unless mentioned specifically. Sample size was determined based on our experience and previous experiments. No data were excluded from analysis. Animal experiments described were repeated with at least three independent replicates with significant results in the same direction as those represented in the figures. All animal studies were randomized in ‘control’ or ‘treated’ groups. However, all animals housed within the same cage were generally placed within the same treatment group. For analysis of tumor samples, identities were blinded from histopathological assessment. All animals were housed in SPF (Specific Pathogen Free) facilities. The animal studies were approved by the Institutional Animal Care Committee of Chungbuk National University.

### 2.2. Adenovirus and Tamoxifen Delivery in Mice

Adenovirus carrying Cre recombinase (*Ad-Cre*) was purchased from Vector Biolabs (Philadelphia, PA, USA). Each mouse was treated with a titer of 2.5 × 10^7^ *Ad-Cre* viral genome copies diluted in 50 μL warm sterile MEM. After the treatment, mice were placed on a warm pad until they woke up. Tamoxifen-containing food was administered every day at 400 mg/kg food (#TD.130860, Teklad diet; ENVIGO, Somerset, NJ, USA). Animal studies were approved by the Institutional Animal Care Committee of Chungbuk National University. Animals were maintained under specific pathogen-free conditions and monitored daily.

### 2.3. KPR Primary Tumor Cell Line Extract

We prepared 6- to 8-week-old *K-Ras^LSL-G12D/+^*, *p53^flox/flox^*, *Runx3^flox/FSF^*, *Tomato**, and *Flp^ERT2^* mice (KPR*^L/F^* mice) and nasally infected them with *Ad-Cre* to induce oncogenic *K-Ras*-dependent lung cancer. Four weeks after the virus infection, we sacrificed the mice and extracted their lungs to obtain the KPR primary tumor cell line. The tumor burden from the lung was sliced into tiny pieces and treated with trypsin-EDTA so that the KPR primary tumor cells were isolated. The cells were stabilized in 20% FBS-containing DMEM for days and cultured in 10% FBS-containing DMEM.

### 2.4. Hematoxylin and Eosin (HE) Staining

H&E staining experiments were performed following standard protocols. Briefly, slides were rehydrated with ethanol, xylene, and water to remove the paraffin. The nuclei were stained with hematoxylin (DAKO, CA, USA, #S3309) for 3 min and the cytoplasm was stained with eosin (Sigma, HT110280) for 30 seconds. Slides were mounted with Permount (Fisher Scientific, SP15-500) after the dehydration and clearing steps.

### 2.5. Histology and Immunohistochemistry

For histological analyses, lungs were inflated with 4% paraformaldehyde or formalin (3.7% formaldehyde) and fixed for 36 h. Fixed paraffin sections were rehydrated, subjected to antigen retrieval, blocked in TBS (0.1% Triton X-100 containing 1% BSA) or DAKO protein-free blocking solution, and sequentially incubated with specific primary antibodies, biotinylated secondary antibodies (DAKO), and the Alexa Fluor system (Invitrogen, Waltham, MA, USA). Terminal deoxynucleotidyl transferase dUTP nick-end labeling (TUNEL) staining was performed using the in situ Cell Death Detection kit (Roche). Images were produced using a conventional microscope mounted with a DP71 digital camera (Olympus), an LSM 710 T-PMT confocal microscope (Carl Zeiss), and an AXIO Zoom.V16 and ApoTome.2 (Carl Zeiss). Images were processed with equivalent parameters using the ZEN Light Edition software (Carl Zeiss) https://www.zeiss.com.cn/microscopy/products/microscope-software/zen-lite/zen-lite-download.html accessed on 6 October 2023.

### 2.6. DNA Exon-Seq Analysis

For the generation of standard exome capture libraries, the Agilent SureSelect Target Enrichment protocol for the Illumina paired-end sequencing library (ver. B.3, June 2015) was used with 1 µg input gDNA. In all cases, the SureSelect Human All Exon V6 or SureSelect Mouse All Exon probe set was used. DNA quantification and DNA quality were analyzed using PicoGreen and agarose gel electrophoresis. One microgram of genomic DNA from each cell line was diluted in EB buffer and sheared to a target peak size of 150–200 bp using the Covaris LE220 focused ultrasonicator (Covaris, Woburn, MA, USA) according to the manufacturer’s recommendations. The 8 microTUBE Strip was loaded into the tube holder of the ultrasonicator, and the DNA was sheared using the following settings: mode, frequency sweeping; duty cycle, 10%; intensity, 5; cycles per burst, 200; duration, 60 sec × 6 cycles; and temperature, 4–7 °C. The fragmented DNA was repaired, an ‘A’ was ligated to the 3′ end, and Agilent adapters were then ligated to the fragments. Once ligation was assessed, the adapter-ligated product was PCR amplified. The final purified product was quantified using the TapeStation DNA screentape D1000 (Agilent, Santa Clara, CA, USA). For exome capture, 250 ng of the DNA library was mixed with hybridization buffers, blocking mixes, RNase block, and 5 µL of SureSelect all-exon capture library, according to the standard Agilent SureSelect Target Enrichment protocol. Hybridization to the capture baits was performed at 65 °C using the heated thermal cycler lid option at 105 °C for 24 h on a PCR machine. The captured DNA was then washed and amplified. The final purified product was quantified using qPCR according to the qPCR Quantification Protocol Guide (KAPA Library Quantification kits for Illumina Sequencing platforms), analyzed using the TapeStation DNA screentape D1000 (Agilent), and sequenced using the HiSeq™ 2500 platform (Illumina, San Diego, CA, USA).

### 2.7. DNA Transfection, IP, and IB

Transient transfections in all cell lines were performed using Lipofectamine Plus reagent and Lipofectamine (Invitrogen). Cell lysates were incubated with the appropriate mono- or polyclonal antibodies (2 μg antibody/500 μg lysate sample) for 3 h at 4 °C, and then with protein G–Sepharose beads (Amersham Pharmacia Biotech, Piscataway, NJ, USA) for 1 h at 4 °C. For detection of endogenous proteins, lysates were incubated with the appropriate mono- or polyclonal antibodies (dilution range 1:1000–1:3000) for 6–12 h at 4 °C, and then with protein G–Sepharose beads (Amersham Pharmacia Biotech) for 3 h at 4 °C. Immunoprecipitates were resolved using SDS–polyacrylamide gel electrophoresis and transferred to a polyvinylidene difluoride membrane (Millipore, Billerica, MA, USA). The membrane was immunoblotted with the appropriate antibodies after blocking and visualized on an Amersham™ Imager 600 (GE Healthcare, Chicago, IL, USA) after treatment with ECL solution (Amersham Pharmacia Biotech).

### 2.8. Antibodies

Antibodies targeting p300 (Cat# sc-584), p53 (Cat# sc-126), Arf (Cat# sc-8340, Cat# sc-53640), Tbp (Cat# sc-421), and Brg-1 (Cat# sc-17796, Cat# sc-10768) were obtained from Santa Cruz Biotechnology (Dallas, TX, USA). Antibodies targeting Brd2 (Cat# H00006046-M01) were obtained from Abnova (Taipei City, Taiwan). All antibodies were diluted to 1:1000. Antibodies targeting Runx3 (Cat# ab40278) were obtained from Abcam (Cambridge, UK) and diluted to 1:3000.

### 2.9. RNA-Seq Analysis

Total RNA was isolated using Trizol reagent (Invitrogen). RNA quality was assessed using an Agilent 2100 bioanalyzer (Agilent Technologies, Amstelveen, The Netherlands) and RNA quantification was performed using an ND-2000 Spectrophotometer (Thermo Inc., DE, USA). Libraries were prepared from total RNA using the NEBNext Ultra II Directional RNA-Seq Kit (NEW ENGLAND BioLabs, Inc., UK). Isolation of mRNA was performed using the Poly(A) RNA Selection Kit (LEXOGEN, Inc., Austria). The isolated mRNAs were used for cDNA synthesis and shearing in accordance with the manufacturer’s instructions. Indexing was performed using the Illumina indexes 1–12. The enrichment step was carried out using PCR. Subsequently, libraries were checked using the TapeStation HS D1000 Screen Tape (Agilent Technologies, Amstelveen, The Netherlands) to evaluate the mean fragment size. Quantification was performed using the library quantification kit and a StepOne Real-Time PCR System (Life Technologies, Inc., Carlsbad, CA, USA). High-throughput sequencing was performed as paired-end 100 sequencing using NovaSeq 6000 (Illumina, Inc., San Diego, CA, USA).

### 2.10. Quantification and Statistical Analysis

Quality control of raw sequencing data was performed using FastQC [26]. Adapters and low-quality reads (<Q20) were removed using FASTX_Trimmer [27] and BBMap [28]. Then, the trimmed reads were mapped to the reference genome using TopHat [29]. The Read Count data were processed based on an FPKM+ Geometric normalization method using EdgeR within R [30]. FPKM (fragments per kb per million reads) values were estimated using Cufflinks [31]. Data mining and graphic visualization were performed using ExDEGA (Ebiogen Inc., Seoul, Republic of Korea). Gene clustering was analyzed using DAVID Bioinformatics Resources 2021 [32].

## 3. Results

### 3.1. Generation of K-Ras^LoxP-Stop-LoxP-G12D/+^, Runx3^flox/FSF^, Tomato*, and Flp^ERT2^ (KR^L/F^) Mice

To determine the roles of *Runx3* in *K-Ras*-activated lung ADCs, we developed a mouse model, *Runx3^Frt-Stop-Frt^* (hereafter *Runx3^FSF^*), in which *Runx3* is deactivated by a *Frt-Stop-Frt* cassette, but can be conditionally restored via deletion of the cassette through the activation of Flippase recombinase (Flp) (Figure 1A and Appendix A). *Runx3^FSF/+^* mice were indistinguishable from *Runx3^+/−^* mice, and *Runx3^FSF/FSF^* mice, similar to conventional *Runx3*^−/*−*^ mice [33], died within 24 h after birth.

*Runx3^FSF/+^* mice were crossed with strains harboring *Rosa26R-Tomato (Tomato*)*, *K-Ras^LoxP-Stop-LoxP-G12D/+^*, *Runx3^flox/flox^*, and *Flp^ERT2^*, yielding *Tomato**, *K-Ras^LoxP-Stop-LoxP-G12D/+^*, *Runx3^flox/FSF^*, and *Flp^ERT2^* mice (KR*^L/F^* mice) (Figure 1B). A *Tomato** allele was included to trace the targeted cells. In these mice, expression of Cre recombinase (Cre) transduced by *Ad-Cre* (adenovirus carrying *CMV* promoter-driven *Cre* recombinase) activates *K-Ras^LoxP-Stop-LoxP-G12D^* and deactivates *Runx3^flox^*, and treatment with tamoxifen (TAM) restores *Runx3* by activating *Flp^ERT2^* (Figure 1B).

### 3.2. Runx3 Restoration Effectively Eliminates Established K-Ras-activated Lung Cancer Cells

To determine whether *Runx3* restoration could regress already-established *K-Ras*-activated lung ADs and/or ADCs, the KR*^L/F^* mice were infected with *Ad-Cre* through nasal inhalation (2.5 × 10^7^ pfu/mouse) [12,25] for *K-Ras* activation and *Runx3* inactivation. Six weeks after *Ad-Cre* infection, the lungs of the mice exhibited Tomato fluorescence under UV light, indicating tumor development (Figure 1C). Microscopic analysis confirmed that the mice developed many lung ADs/ADCs (Figure 1C). The remaining mice were fed normal food (KR*^L/F^*-TAM(-), *n* = 5) or tamoxifen-containing food (KR*^L/F^*-TAM(+), *n* = 5) to promote *Runx3* restoration. KR*^L/F^*-TAM(-) mice survived for an average of 13.2 weeks after infection, and all died by 14 weeks after infection (Figure 1D). In contrast, all KR*^L/F^*-TAM(+) mice survived for an average of 28.2 weeks after infection, and all died by 30 weeks after infection (Figure 1D). These results demonstrate that *Runx3* restoration extends the survival of lung-cancer-induced KR*^L/F^* mice by 15 weeks.

In a parallel experiment, KR*^L/F^*-TAM(-) and KR*^L/F^*-TAM(+) mice were sacrificed at 4 or 10 weeks after tamoxifen treatment (Figure 1D). KR*^L/F^*-TAM(-)-4w mouse lungs exhibited high levels of Tomato fluorescence under UV light and developed large lung ADs/ADCs (Figure 1E). However, the lungs of KR*^L/F^*-TAM(+)-4w mice exhibited very low levels of Tomato fluorescence under UV light (Figure 1F). The number of Tomato-positive cells did not increase until 10 weeks after *Runx3*-restoration (Figure 1G). Enlarged microscopic images of the figure are shown in Appendix A. Once the *Rosa26R-Tomato* allele is targeted, Tomato fluorescence is maintained throughout the lifespan of the targeted cells. Therefore, the rare Tomato-positive cells detected in the lungs of KR*^L/F^*-TAM(+)-4w and KR*^L/F^*-TAM(+)-10w mice suggest that nearly all the *K-Ras*-activated lung AD cells and ADC cells were eliminated through *Runx3* restoration. Genotyping of the lung tumors confirmed *K-Ras* activation, *Runx3* inactivation by *Ad-Cre* infection, and *Runx3* restoration via tamoxifen treatment (Appendix A).

### 3.3. Runx3 Restoration Eliminates K-Ras-activated Lung Cancers by Inducing Apoptosis

To elucidate the mechanism by which *Runx3* restoration eliminated *K-Ras*-activated lung ADs and ADCs, we obtained lungs from *Ad-Cre*-infected KR*^L/F^* mice 1 week after tamoxifen treatment. Tomato and terminal deoxynucleotidyl transferase dUTP nick-end labeling (TUNEL) staining of the mouse lungs showed that the Tomato-positive lung cancer cells of KR*^L/F^*-TAM(-)-1w mice were TUNEL-negative. However, most of the Tomato-positive lung cancer cells of KR*^L/F^*-TAM(+)-1w mice were TUNEL-positive (Figure 2A), indicating that the *Runx3*-restored cells underwent apoptosis. Enlarged microscopic images of the figure are shown in Appendix A. We previously reported that *Arf* is a major target of Runx3 [20]; consistently, *Runx3* restoration induced Arf and p53 expression in the lung cancers (Figure 2A). Enlarged microscopic images of the figure are shown in Appendix A. Consistent with this, the lungs of KR*^L/F^*-TAM(+)-4w mice contained a few lesions that were undergoing regression, and the cells in these lesions were TUNEL-positive (Figure 2B,C). These results demonstrate that *Runx3* restoration activates the Arf-p53 pathway and eliminates *K-Ras*-activated lung cancer cells by inducing apoptosis.

### 3.4. K-Ras-Activated Lung Cancer Began to Recur at 14 Weeks after Runx3 Restoration

All of the *Runx3*-restored mice began to die at 20 weeks after tamoxifen treatment (26 weeks after *Ad-Cre* infection) (Figure 1D). In a parallel experiment, KR*^L/F^*-TAM(+) mice were sacrificed at 14 weeks after tamoxifen treatment (Figure 1D). Analysis of the KR*^L/F^*-TAM(+)-14w mouse lungs showed that small ADCs had developed in all four mice (Figure 3A). The ADCs were Tomato-positive, suggesting that the cancers recurred from remnant *K-Ras*-activated cells (Figure 3A). Whole-exon sequencing indicated that *K-Ras* was mutated (*K-Ras^G12D^*) in KR*^L/F^*-TAM(-)-0w mouse lung ADCs and recurrent lung ADCs, as targeted in the *K-Ras^LSL-G12D^* allele. There were no additional mutations in *K-Ras*, and none of the other known major oncogenes (*Egfr*, *B-Raf*, *Alk*, *Mek*, *Stk11*, *Smarca4*, and *Pi3ka*) or tumor suppressors (*Rb1*, *Apc*, and *p53*) involved in lung cancer were mutated in any of the ADCs (Figure 3B and Appendix A). This suggests that secondary oncogene activation was not involved in the cancer recurrence.

Genotyping of the recurrent lung ADCs confirmed that the Stop cassette was removed from the *Runx3^FSF^* allele by tamoxifen-activated *Flp^ERT2^* (Figure 3C). However, immunostaining showed that Runx3 expression was considerably lower in the majority of the recurrent ADCs than in the adjacent normal region (Figure 3D). Methylation-specific PCR (MS-PCR) showed that the CpG island of *Runx3* was hyper-methylated in six of eight recurred ADCs (Figure 3E). These results suggest that the restored *Runx3* allele was spontaneously inactivated, mainly through DNA hyper-methylation in the recurred lung ADCs. Therefore, it is likely that the quiescent *K-Ras*-activated cells that remained after *Runx3* restoration re-established lung tumors due to spontaneous silencing of the restored *Runx3* allele. The nucleotide sequence of the *Runx3* CpG island subjected to MS-PCR and the PCR primers used are shown in Appendix A.

### 3.5. Runx3 Inactivation Is Essential for the Maintenance of K-Ras-Activated Lung Cancer

*K-Ras^LA1/+^* mice, a mouse strain carrying oncogenic alleles of *K-Ras* that can be activated by a spontaneous recombination event, develop a range of tumor types, predominantly lung cancer [16]. To understand the role of *Runx3* in lung tumorigenesis activated by *K-Ras* alone, we crossed *K-Ras^LA1/+^* mice with *Runx3^FSF/+^* mice and *Flp^ERT2^* mice, yielding *K-Ras^LA1/+^*; *Runx3^+/+^* (K*^LA1^*R^+/+^) and *K-Ras^LA1/+^; Runx3^FSF/+^*; and *Flp^ERT2^* (K*^LA1^*R*^FSF/+^*) mice (Figure 4A). In the K*^LA1^*R*^FSF/+^* mice, one allele of *Runx3* is wild type and the other allele (*Runx3^FSF^*) is deactivated by the *Frt-Stop-Frt* cassette. The *Runx3^FSF^* allele can be restored by tamoxifen, which activates *Flp^ERT2^*.

We measured the lifespan of K*^LA1^*R*^+/+^* mice and K*^LA1^*R*^FSF/+^* mice in the absence of tamoxifen. The K*^LA1^*R*^+/+^* mice (*n* = 15) developed lung cancer and began to die at 26 weeks after birth, and all the mice died within 72 weeks after birth (average lifespan, 51.7 weeks) (Figure 4B,C). The K*^LA1^*R*^FSF/+^* mice (*n* = 18) began to die at 10 weeks after birth, and all the mice died within 70 weeks after birth (the average lifespan was 39.5 weeks, which is 12.2 weeks shorter than that of K*^LA1^*R*^+/+^* mice) (Figure 4B,C). These results demonstrate that inactivation of one allele of *Runx3* significantly shortened the survival of *K-Ras^LA1/+^* mice (*p* = 0.04).

In a parallel experiment, K*^LA1^*R*^FSF/+^* mice were fed tamoxifen-containing food for *Runx3* restoration for 2 weeks starting at 10 weeks after birth (K*^LA1^*R*^FSF/+^*-TAM(+), *n* = 15). *Runx3* restoration significantly extended the survival of the K*^LA1^*R*^FSF/+^* mice: K*^LA1^*R*^FSF/+^*-TAM(+) mice survived for an average of 52.9 weeks after birth, which is 13.4 weeks longer than the K*^LA1^*R*^FSF/+^*-TAM(-) mice (*p* = 0.02) (Figure 4B,C). The survival curve and the average lifespan of K*^LA1^*R*^FSF/+^*-TAM(+) mice were similar to that of K*^LA1^*R*^+/+^* mice (Figure 4B,C). MS-PCR analysis of the lung ADCs of the K*^LA1^*R*^FSF/+^*-TAM(-) mice and K*^LA1^*R*^FSF/+^*-TAM(+) mice revealed that *Runx3* was silenced by CpG island hyper-methylation in all the analyzed lung ADCs (Figure 4D,E). Immunostaining analysis confirmed that the level of Runx3 was considerably lower in ADC cells than in adjacent normal cells (Figure 4F). These results are consistent with the previous observation that *Runx3* is silenced by CpG island hyper-methylation in nearly all the lung ADCs activated by *K-Ras* alone [25].

If *Runx3* inactivation is essential for the maintenance of *K-Ras*-activated lung cancer, the survival of the model mice for activation via *K-Ras* alone should be directly related to the number of functional *Runx3* alleles. K*^LA1^*R*^+/+^* mice have two functional *Runx3* alleles. K*^LA1^*R*^FSF/+^*-TAM(-) mice have only one functional *Runx3* allele because the other allele is inactivated. K*^LA1^*R*^FSF/+^*-TAM(+) mice have two functional *Runx3* alleles because the inactivated *Runx3* allele is restored. The survival of the mouse models was indeed directly related to the number of functional *Runx3* alleles (Figure 4B,C). These results confirm that *Runx3* inactivation is essential for the maintenance of lung cancer activated by *K-Ras* alone.

The expression of K-Ras*^G12D^* in the lung ADCs developed in K*^LA1^*R*^+/+^*, K*^LA1^*R*^FSF/+^*-TAM(-), and K*^LA1^*R*^FSF/+^*-TAM(+) mice was confirmed through immunoblotting (IB) using a K-Ras*^G12D^*-specific antibody (Figure 4G). Restoration of *Runx3* in the lung ADCs developed in K*^LA1^*R*^FSF/+^*-TAM(+) mice was confirmed with genomic DNA PCR (Figure 4H and Appendix A).

### 3.6. The Tumor-Suppressive Activity of Runx3 Is Largely Dependent on p53

Next, we investigated whether activation of the Arf-p53 pathway is essential for the regression of *K-Ras*-activated lung cancers induced via *Runx3* restoration. For this purpose, we crossed *p53^flox/flox^* mice with KR*^L/F^* mice and obtained *K-Ras^LSL-G12D/+^*, *p53^flox/flox^, Runx3^flox/FSF^*, *Tomato**, and *Flp^ERT2^* mice (KPR*^L/F^* mice). In KPR*^L/F^* mice, *Ad-Cre* infection activated *K-Ras^LSL-G12D^*, deactivated *p53^flox^* and *Runx3^flox^*, and labeled the targeted cells with Tomato fluorescence (Figure 5A). Treatment with tamoxifen restored *Runx3* by deleting the *Frt-Stop-Frt* cassette from the *Runx3^FSF^* allele via activation of *Flp^ERT2^* (Figure 5A).

Two weeks after *Ad-Cre* infection, KPR*^L/F^* mice developed lung cancer (Figure 5B). Microscopy revealed that the cancers that developed in the KPR*^L/F^* mice showed nuclear pleomorphism with prominent nucleoli and scattered cancer giant cells, as well as more advanced histopathology than that of cancers developed in KR*^L/F^* mice (Figure 5C). The *Ad-Cre*-infected KPR*^L/F^* mice began to die at 8 weeks after *Ad-Cre* infection, and all the mice died within 11 weeks (median survival, 9.8 weeks) (Figure 5D). The median survival of KR*^L/F^* mice was 13.2 weeks (Figure 1D and Figure 5D). These results indicate that the lifespan of KPR*^L/F^* mice was approximately 3.4 weeks shorter than that of KR*^L/F^* mice (*p* = 0.0018).

After confirming tumor development in KPR*^L/F^* mice (2 weeks after *Ad-Cre* infection, Figure 5B), the mice were fed normal (KPR*^L/F^*-TAM(-)) or tamoxifen-containing food (KPR*^L/F^*-TAM(+)) for 2 weeks (Figure 5E). The median survival was 9.8 weeks in KPR*^L/F^*-TAM(-) mice and 14.2 weeks in KPR*^L/F^*-TAM(+) mice (Figure 5E). These results demonstrate that *Runx3* restoration extended the survival of KPR*^L/F^* mice by 4.4 weeks (p = 0.0004). Although *Runx3* restoration significantly extended the survival of KPR*^L/F^* mice (≈4.4 weeks), a comparison of the effect of *Runx3* restoration with that on KR*^L/F^* mice (>18 weeks) (Figure 1D) indicates that the tumor-suppressive activity of Runx3 is largely dependent on p53.

Genotyping of the KPR*^L/F^*-TAM(-) and KPR*^L/F^*-TAM(+) lung cancers confirmed targeting of *K-Ras^LSL-G12D^*, *p53^flox^*, and *Runx3^flox^* alleles by *Ad-Cre* infection (Appendix A). We also confirmed that the *Frt-Stop-Frt* cassette was deleted from the *Runx3^FSF^* allele in the KPR*^L/F^*-TAM(+) cancers, indicating that *Runx3* was restored in these cancers (Appendix A).

### 3.7. Runx3 Restoration Recovers the R-Point and Induces Arf Expression

We previously reported that RUNX3 plays a key role in R-point regulation by forming the R-point-associated RUNX3-containing activator (Rpa-RX3-AC) complex, which induces ARF expression in response to oncogenic K-RAS activity [20]. The Rpa-RX3-AC complex includes RUNX3, p300, BRD2, MLLs, the chromatin remodeling complex (SWI/SNF), and the basal transcription machinery (TFIID). In normal cells, the Rpa-RX3-AC complex is formed only for short intervals (1–2 h) when RAS is activated by mitogenic signals and quickly dissociates as the signal attenuates [20]. Then, the cells undergo cell cycle progression. However, in oncogenic RAS-expressing cells, the Rpa-RX3-AC complex is maintained for a long time, as the oncogenic RAS signal is not attenuated. Thus, the Rpa-RX3-AC complex selectively activates the ARF-p53 pathway in response to the aberrant persistence of the RAS signal. However, whether the ARF-p53 pathway is sensitive enough to respond to a persistent low level of oncogenic K-RAS activity originating from heterozygous oncogenic *K-RAS* mutation remains unclear [13,14,15]. To determine whether the pathway responds to heterozygous oncogenic *K-Ras* mutation, we measured the expression of Arf in KPR*^L/F^*-TAM(-) and KPR*^L/F^*-TAM(+) lung cancers, which bear a heterozygous oncogenic *K-Ras* mutation. The results of immunostaining showed that Arf was not expressed in KPR*^L/F^*-TAM(-) lung cancers, whereas it was expressed in KPR*^L/F^*-TAM(+) lung cancers in which *Runx3* was restored (Appendix A). These results suggest that the Arf-p53 pathway is inactivated in the absence of *Runx3*, whereas it is activated through the restoration of *Runx3* in response to heterozygous oncogenic *K-Ras* mutation.

To determine whether the Rpa-RX3-AC complex formation is also sensitive enough to respond to the heterozygous oncogenic *K-Ras* mutation, we obtained immortalized cell lines from KPR*^L/F^*-TAM(-) lung cancers (KPR^-^) (Figure 5F). Treatment of KPR^-^ cell lines with 4-hydroxytamoxifen (4-OHT) restored *Runx3*, generating KPR^restored^ cell lines (Figure 5F and Appendix A). We confirmed that *Runx3* was restored at 8 h after 4-OHT treatment (Figure 5G). Immunoprecipitation (IP) followed by IB showed that the restored Runx3 associated with p300, Brd2, Brg-1 (a component of SWI/SNF), and Tbp (a component of TFIID), indicating the formation of the Rpa-RX3-AC complex (Figure 5G). The Rpa-RX3-AC complex was maintained until 24 h after 4-OHT treatment. Arf was not expressed in KPR^-^ cells, whereas it was expressed in *Runx3*-restored cells (KPR^restored^) (Figure 5G). These results demonstrate that the Rpa-RX3-AC complex is sensitive enough to activate the Arf-p53 pathway in response to a persistent low level of oncogenic K-Ras activity.

The KPR^-^ and KPR^restored^ cell lines were transfected with empty vector or p53-expressing plasmid. Ectopic expression of p53 in the KPR^-^ cells, in which Arf was not expressed, activated Caspase-3 weakly (Figure 5H). Caspase-3 was also weakly activated in the KPR^restored^ cells, in which *p53* was deleted and *Arf* was induced through *Runx3* restoration (Figure 5H). However, expression of p53 in KPR^restored^ cells strongly activated Caspase-3 (Figure 5H). Densitometric analysis of the band intensities revealed that the combination of *Runx3* restoration and p53 expression resulted in an approximately eight-fold stronger activation of Caspase-3 than either *Runx3* restoration or p53 expression alone (Appendix A). These results are consistent with our in vivo observations that the tumor-suppressive activity of Runx3 is largely dependent on p53 activity (Figure 2B and Figure 5D). The results suggest that the R-point-associated Arf-p53 pathway is abrogated with *Runx3* inactivation and recovered with *Runx3* restoration in lung cancer cells bearing a heterozygous oncogenic *K-Ras* mutation.

### 3.8. Runx3 Restoration Inhibits Proliferation of K-Ras-Activated Lung Tumor Cells in a p53-Independent Manner

Although the tumor-suppressive activity of Runx3 was largely dependent on p53, *Runx3* restoration extended the survival of KPR*^L/F^* mice (Figure 5E). Microscopy revealed that the lung cancers developed in KPR*^L/F^*-TAM(-) mice and KPR*^L/F^*-TAM(+) mice were pathologically indistinguishable (Appendix A). However, the proliferation rate of KPR^restored^ cells was lower than that of KPR^-^ cells (*p* = 0.003) (Figure 6A). Consistently, the number of PCNA-positive cells in KPR*^L/F^* lung tumors was significantly reduced with *Runx3* restoration (KPR*^L/F^*-TAM(-) = 797.2/mm^2^; KPR*^L/F^*-TAM(+) = 423.5/mm^2^, *p* = 0.023) (Figure 6B,C). Enlarged microscopic images of Figure 6B are shown in Appendix A. These results suggest that *Runx3* restoration inhibits the proliferation of *K-Ras*-activated lung cancer cells.

To identify genes regulated by *Runx3* in the *K-Ras*-activated lung cancer cells, we performed mRNA sequencing (RNA-seq) in KPR^-^ cells and KPR^restored^ cells. Analysis of the Z-scores revealed that 3,194 and 3,002 genes were induced and suppressed, respectively, in response to *Runx3* restoration (Figure 6D). Major signaling pathways upregulated with *Runx3* restoration involved apoptosis and negative regulation of proliferation (Figure 6E). On the other hand, genes involved in the positive regulation of cell proliferation and DNA replication were suppressed with RUNX3 expression (Figure 6F). *RUNX3*-dependent up and downregulated genes involved in apoptosis and negative regulation of proliferation are listed in Appendix A. Although *Runx3* restoration upregulated the expression of many genes involved in apoptosis, the KPR^restored^ cells did not undergo apoptosis (Figure 5H), suggesting that the Arf-p53 pathway is essential for inducing apoptosis in *K-Ras*-activated lung cancer cells. Taken together, these results suggest that *Runx3* restoration suppresses *K-Ras*-activated lung cancer mainly through the activation of Arf-p53 pathway-mediated apoptosis and partly through p53-independent inhibition of proliferation. Detailed RNA-seq results are provided in the Excel file (Appendix A). Further research is required to identify statistically meaningful target genes of the Runx3-induced p53-independent inhibition of proliferation.

## 4. Discussion

Rapid recurrence of cancer after treatment with oncogenic *K-RAS*-specific inhibitors suggests that early tumor lesions are resistant to oncoprotein inhibitors, and that secondary oncogene activation and resistance to the effect of the inhibitors leads to the resumption of cell proliferation [34]. ADs that develop without oncogene activation should be resistant to oncoprotein inhibitors. Therefore, to develop new treatment strategies against *K-RAS*-activated lung cancer with a durable response, it is necessary to understand how ADs develop. The ARF-p53 pathway is an effective defense mechanism against oncogenic *K-RAS* mutations [9,35]. Therefore, the development of *K-RAS*-activated lung ADs must be accompanied by the inactivation of the ARF-p53 pathway. However, oncogenic *K-Ras*-mutated cells develop into lung AD in the absence of p53 mutation [16,36,37]. In addition, in a *K-Ras*-activated mouse lung cancer model, *p53* restoration eliminates only ADCs, leaving ADs intact [13,14]. These results led to the speculation that the Arf-p53 pathway has inherent limits in its capacity to respond to heterozygous oncogenic *K-Ras* mutation, and, therefore, oncogenic *K-Ras* alone is sufficient to induce lung ADs [13,14,15].

However, another possibility was suggested. The initial step of colorectal AD development is the inactivation of adenomatous polyposis coli (APC), and activation of *K-Ras* occurs after AD development [18,38]. In addition, *Apc* restoration in established *Apc*-inactivated and *K-Ras*-activated mouse colorectal ADCs drives rapid and widespread cancer cell differentiation and sustained regression without recurrence [39]. These results indicate that mammals have evolved an effective defense mechanism against oncogenic *K-Ras* mutations, and the mechanism is abrogated in colorectal cancer through *Apc* inactivation, which induces the formation of colorectal ADs. Lung cancers develop through a similar multistep tumorigenesis pathway (ADs progress into ADCs). We previously reported that inactivation of *Runx3* in the mouse lung induces the development of ADs [24,25]. In addition, *Runx3* inactivation is an earlier event than *K-Ras* activation in a carcinogen-induced mouse lung cancer model that recapitulates the features of *K-RAS*-driven human lung cancers [40]. Runx3 plays a key role in the R-point decision-making machinery, which senses aberrant oncogenic signals and activates the Arf-p53 pathway [20]. Therefore, a single molecular event, the inactivation of *Runx3*, results in both AD development and the disruption of the Arf-p53 pathway. Indeed, *K-Ras* activation, with or without *p53* inactivation, in an extremely small number of cells failed to induce pathologic lesions for up to 1 year [40]. In contrast, *Runx3* inactivation and *K-Ras* activation with the same targeting method led to the rapid induction of lung ADs and ADCs, and it caused lethality in all the targeted mice within 3 months [40]. Therefore, *Runx3* restoration may regress *K-Ras*-mutated lung tumors and result in sustained regression without recurrence, similar to the effect of *Apc* restoration. In this study, *Runx3* restoration in a mouse lung cancer model regressed *K-Ras*-activated ADs as well as ADCs, suppressed secondary oncogene activation, and markedly extended the survival of mice (by approximately 15 weeks). Knockdown of oncogenic *K-Ras* regresses mouse lung cancer; however, the lung cancer recurs after 2 weeks with secondary oncogene activation [8]. In this study, *Runx3*-restored mouse lung cancer did not recur until 10 weeks after regression (14 weeks after *Runx3* restoration). The recurred tumor demonstrated spontaneous inactivation of the restored *Runx3* allele without secondary oncogene activation. These results show that expression of Runx3 could be helpful for the treatment of lung cancer and for achieving sustained regression.

Arf expression in *K-Ras*-activated lung cancer was stopped upon *Runx3* inactivation and recovered with *Runx3* restoration (Figure 5G and Appendix A). The tumor-suppressive activity of Runx3 was largely dependent on p53 activity, although not completely (Figure 5D). Taken together, these results suggest that *Runx3* is an essential upstream regulator of Arf-p53 pathway activation. These observations might explain the rapid recurrence of *K-Ras*-induced lung cancers with secondary oncogene activation after initial regression due to *K-Ras* suppression: inhibition of oncogenic K-Ras causes the cancer to regress; however, *K-Ras* mutation-free AD cells are resistant to the inhibition and are cancer-prone because their Arf-p53 pathway (oncogene surveillance mechanism) is suppressed through *Runx3* inactivation. In contrast, *Runx3* restoration recovers the oncogene surveillance mechanism and could, therefore, lead to inhibition of secondary oncogene activation as well as cancer regression. It has long remained unclear why the Arf-p53 pathway fails to eliminate *K-Ras*-activated lung ADs [13,14]. The present results suggest that p53 restoration failed to regress lung ADs not because the Arf-p53 pathway had an inherent limitation in responding to oncogenic K-Ras activity, but because the pathway was disrupted by *Runx3* silencing in lung ADs. This does not explain why the p53 mutation still occurs after *K-RAS* activation in lung tumorigenesis. The ARF-p53 pathway protects cells from oncogene activation. In contrast, the ATM/ATR-p53 pathway protects cells from genome instability [41]. Therefore, p53 mutations at relatively late stages of lung tumorigenesis may be associated with disruption of the ATM/ATR-p53 pathway-mediated defense against genome instability.

## 5. Conclusions

Not only in *K-RAS*-activated lung cancers, but in almost all other malignancies, clinical responses have been yielded through the application of targeted therapies that inhibit activated oncogenes, but despite the application of these therapies, tumor recurrence has eventually resulted [42,43]. Therefore, it seems to be of therapeutic value to identify tumor suppressor pathways capable of regressing established cancers and inhibiting cancer recurrence. This study identified *Runx3* as such a tumor suppressor that could effectively regress established *K-Ras*-activated mouse lung cancer and inhibit cancer recurrence.

## Figures and Tables

**Figure 1 cells-12-02438-f001:**
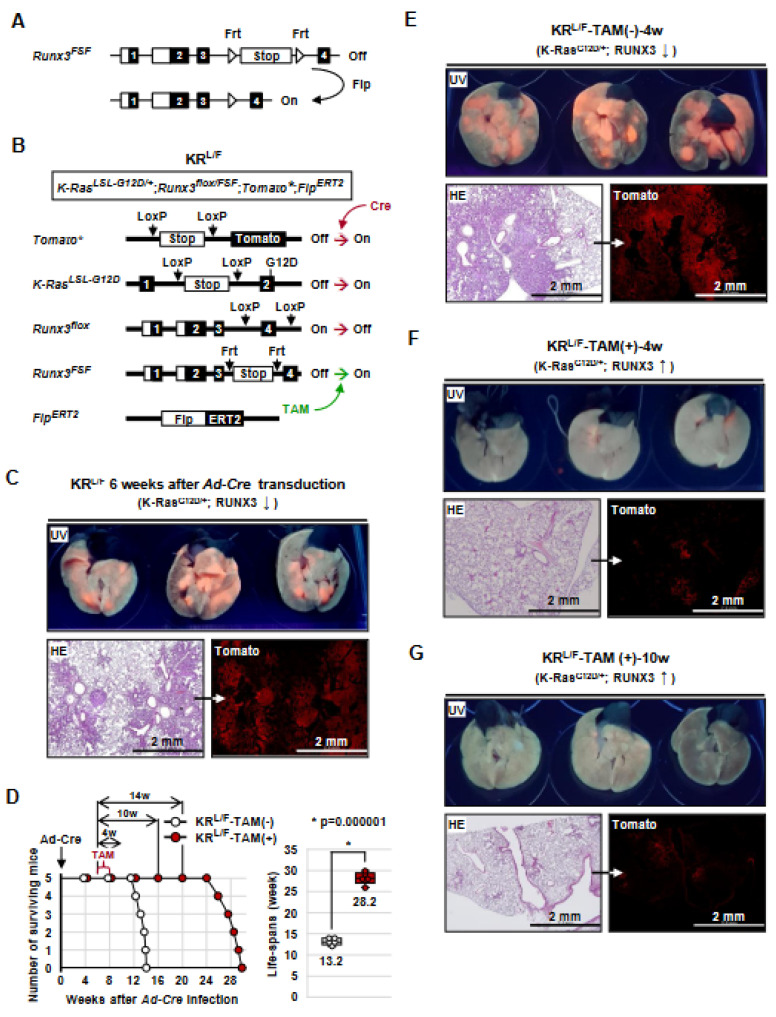
*Runx3* restoration regresses *K-Ras*-dependent lung tumors. (**A**) Schematic representation of the structure of the *Runx3^Frt-Stop-Frt^* (*Runx3^FSF^*) allele. The *Runx3^FSF^* allele is deactivated by the *Frt-Stop-Frt* cassette and restored by Flippase (Flp) recombinase. (**B**) Schematic representation of the structures of the *Rosa26R-Tomato* (Tomato*), *K-Ras^LoxP-Stop-LoxP-G12D^* (*K-Ras^LSL-G12D^*), *Runx3^flox^*, *Runx3^FSF^*, and *Flp^ERT2^* alleles of KR^L/F^ mice. Cre recombinase activates *K-Ras* by removing a knocked-in Stop transcriptional cassette from the *K-Ras^LoxP-Stop-LoxP-G12D^* allele and inactivates the *Runx3^flox^* allele by deleting exon 4. Treatment with tamoxifen (TAM) activates Flippase-ERT2 (*Flp^ERT2^*) recombinase, leading to the restoration of *Runx3* via removal of a knocked-in *Frt-Stop-Frt* cassette from the *Runx3^FSF^* allele. (**C**) Gross images of Tomato fluorescence emitted under UV light from the lungs of KR^L/F^ mice (6 weeks after *Ad-Cre* infection) and microscopic images of the lungs stained with HE (left) or anti-Tomato antibody (right). (**D**) Survival curves of the *Ad-Cre*-infected KR*^L/F^* mice. Six weeks after *Ad-Cre* infection, the mice were fed normal food (KR*^L/F^*-TAM(-) group, *n* = 5) or tamoxifen-containing food (KR*^L/F^*-TAM(+) group, *n* = 5) for two weeks, followed by normal food in all mice. The median survival of the KR*^L/F^*-TAM(-) group and the KR*^L/F^*-TAM(+) group were 13.2 weeks and 28.2 weeks, respectively (*p* = 0.000001). (**E**) Gross images of Tomato fluorescence emitted under UV light from the lungs of KR*^L/F^*-TAM(-)-4w mice (control mice, Figure 2B) and microscopic images of the lungs stained with HE (left) or anti-Tomato antibody (right). (**F**,**G**) Gross images of Tomato fluorescence emitted under UV light from the lungs of KR*^L/F^*-TAM(+)-4w mice and KR*^L/F^*-TAM(+)-10w mice (fed tamoxifen-containing food, Figure 2B), and microscopic images of the lungs stained with HE (left) or anti-Tomato antibody (right).

**Figure 2 cells-12-02438-f002:**
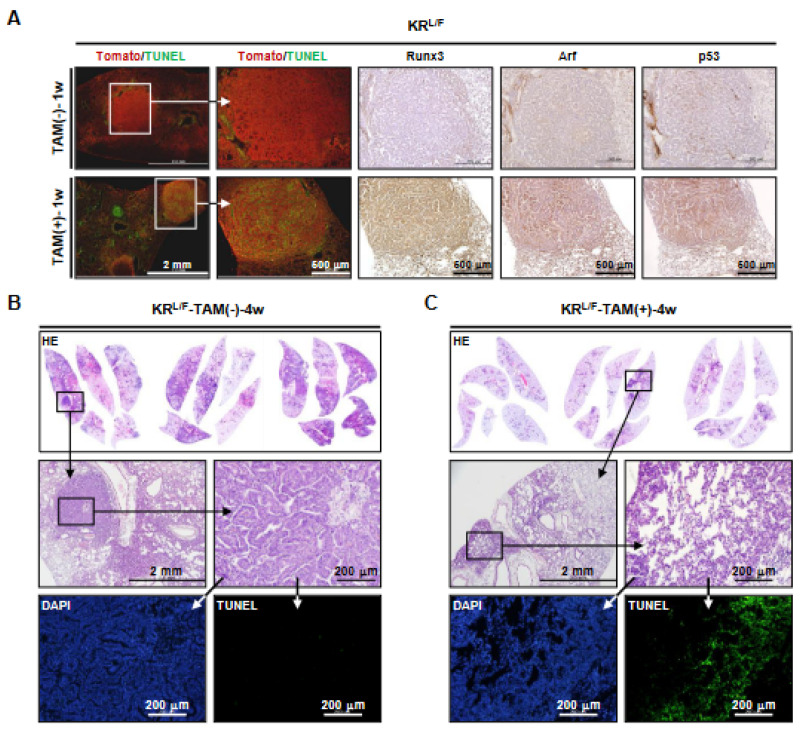
*Runx3* restoration regresses *K-Ras*-activated lung cancers by inducing apoptosis. (**A**) *Ad-Cre*-infected KR*^L/F^* mice were fed normal food or tamoxifen-containing food for 1 week (KR*^L/F^*-TAM(-)-1w and KR*^L/F^*-TAM(+)-1w, respectively). Microscopic images of mouse lungs subjected to Tomato and TUNEL staining are shown. Microscopic images of the adjacent sections stained with anti-Runx3, anti-Arf, and anti-p53 are shown on the right. (**B**,**C**) Microscopic images of KR*^L/F^*-TAM(-)-4w and KR*^L/F^*-TAM(+)-4w mouse lungs subjected to HE and TUNEL staining.

**Figure 3 cells-12-02438-f003:**
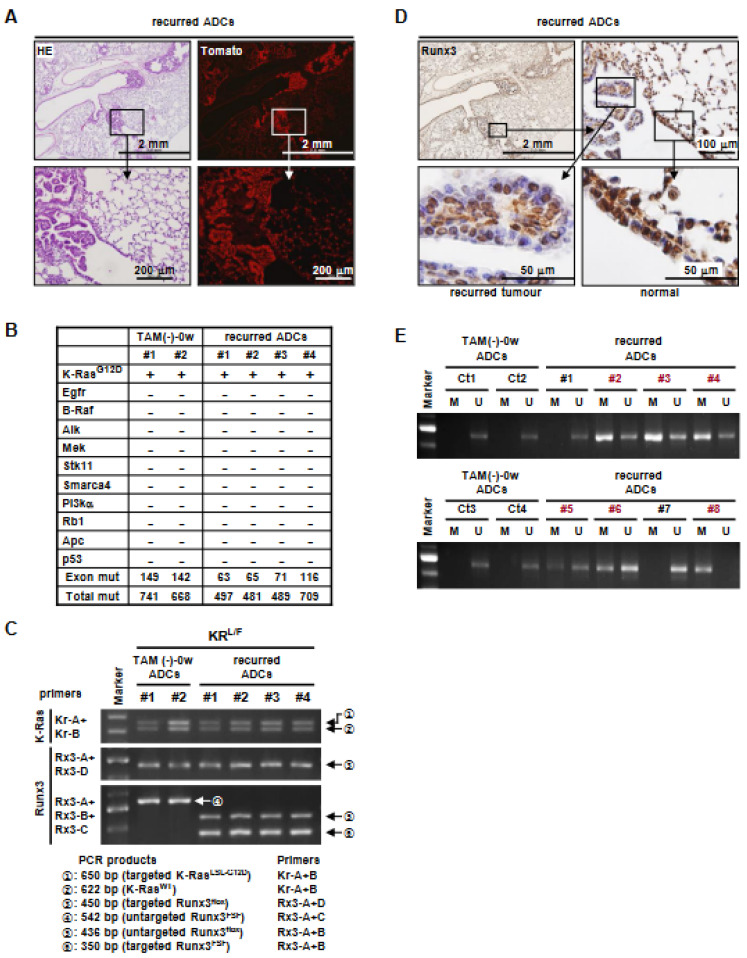
Lung cancers regressed by *Runx3* restoration recur with silencing of the restored *Runx3* allele. (**A**) Microscopic images of KR*^L/F^*-TAM(+)-14w mouse lungs (Figure 2B) subjected to HE and Tomato staining. Magnified images of the boxed regions are shown below. (**B**) Two control lung ADCs of KR*^L/F^*-TAM(-)-0w and four recurred lung ADCs of KR*^L/F^*-TAM(+)-14w mice were analyzed via whole-exon sequencing. Among the known major oncogenes involved in lung cancer, only *K-Ras^G12D^* mutation was detected. Exon mut, number of mutations detected within exons; Total mut, number of mutations detected within the genome. (**C**) Targeting of the *K-Ras^LSL-G12D^, Runx3^flox^*, and *Runx3^FSF^* alleles by *Ad-Cre* infection followed by tamoxifen treatment in cancers was verified through genomic PCR. (**D**) Runx3 expression detected with anti-Runx3 antibody (1E10) in lung ADCs developed in KR*^L/F^*-TAM(+)-14w mice. Magnified images of the boxed regions are shown. (**E**) DNA methylation of the *Runx3* CpG island detected using MS-PCR in lung ADCs developed in KR*^L/F^*-TAM(-)-0w mice and four recurred lung ADCs of KR*^L/F^*-TAM(+)-14w mice. *Runx3*-inactivated ADCs produced via DNA methylation are indicated with red letters. M, methylated *Runx3* CpG island; U, unmethylated *Runx3* CpG island.

**Figure 4 cells-12-02438-f004:**
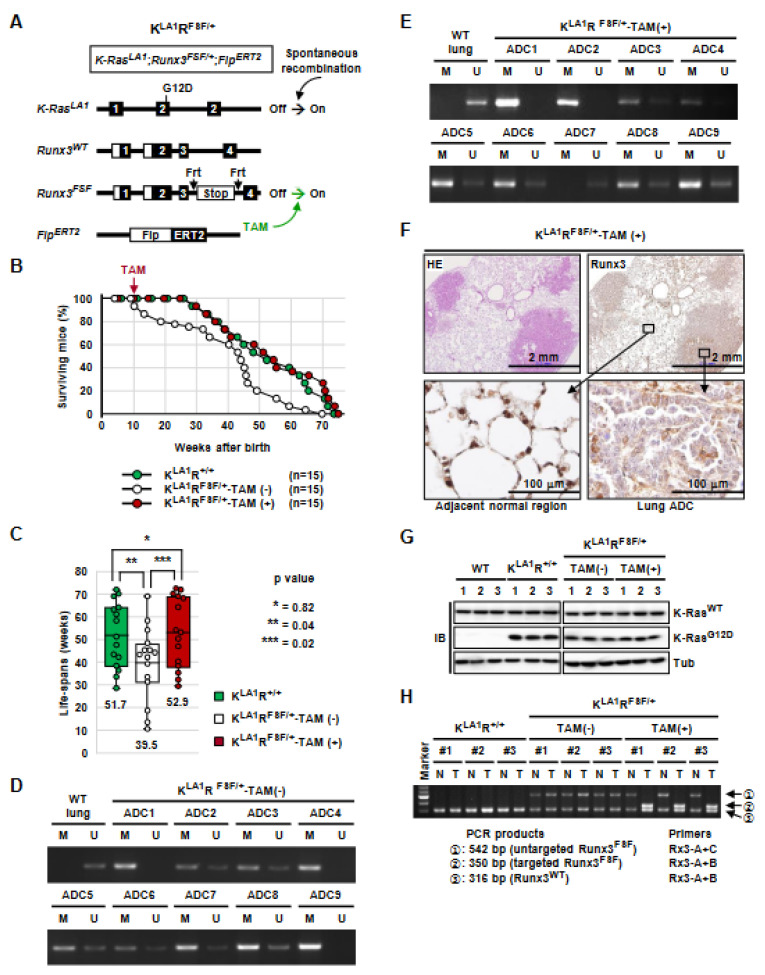
*Runx3* inactivation is essential for the maintenance of *K-Ras*-activated lung cancer. (**A**) Schematic representation of the experimental strategy used for examining the role of *Runx3* in the maintenance of mouse lung cancer induced through *K-Ras*-activation alone. The K*^LA1^*R*^FSF/+^* mice bear the *K-Ras^LA1/+^*, *Runx3^FSF/+^*, and *Flp^ERT2^* alleles. In the K*^LA1^*R*^FSF/+^* mice, *K-Ras* was activated through spontaneous recombination. Treatment with tamoxifen (TAM) restored one allele of *Runx3* by activating *Flp^ERT2^*, which deleted the *Frt-Stop-Frt* cassette from the *Runx3^FSF^* allele. (**B**) Survival curves of K*^LA1^*R*^+/+^* mice and K*^LA1^*R*^FSF/+^* mice infected with *Ad-Cre*. Ten weeks after birth, the K*^LA1^*R*^FSF/+^* mice were fed normal food (K*^LA1^*R*^FSF/+^*-TAM(-), *n* = 15) or tamoxifen-containing food (K*^LA1^*R*^FSF/+^*-TAM(+), *n* = 15) for two weeks, followed by normal food in all the mice. (**C**) Statistical analysis of the lifespan of K*^LA1^*R*^+/+^* mice and K*^LA1^*R*^FSF/+^* mice treated with or without tamoxifen. (**D**,**E**) DNA methylation of the *Runx3* CpG island detected through MS-PCR in lung ADCs developed in 50-week-old K*^LA1^*R*^FSF/+^*-TAM(-) mice and K*^LA1^*R*^FSF/+^*-TAM(+) mice. M, methylated *Runx3* CpG island; U, unmethylated *Runx3* CpG island. (**F**) Runx3 expression detected with anti-Runx3 antibody (1E10) in lung ADCs developed in 50-week-old K*^LA1^*R*^FSF/+^*-TAM(+) mice. Magnified images of the boxed regions are shown. (**G**) Spontaneous activation of the *K-Ras^LA1/+^* allele in lung ADCs developed in K*^LA1^*R*^+/+^*, K*^LA1^*R*^FSF/+^*-TAM(-), and K*^LA1^*R*^FSF/+^*-TAM(+) mice was confirmed via immunoblotting with anti-K-Ras*^G12D^* antibody. (**H**) The restoration of the *Runx3^FSF^* allele in ADCs developed in K*^LA1^*R*^FSF/+^*-TAM(+) mice was verified through genomic PCR. Band ② indicates restoration of *Runx3*. The sizes of PCR products and applied primers are shown. Schematic depictions of the alleles before or after targeting with *Ad-Cre* or tamoxifen, along with the predicted sizes of the PCR products, are shown in Appendix A. N = normal tissue (tail) before tamoxifen treatment, T = lung ADC.

**Figure 5 cells-12-02438-f005:**
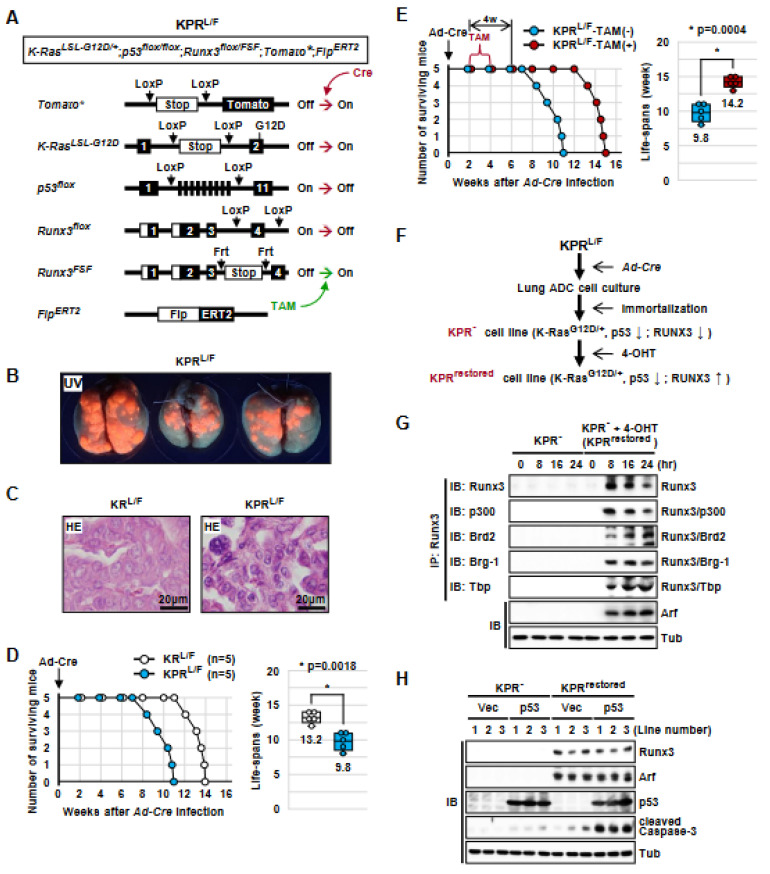
*Runx3* restoration eliminates lung cancers through the Arf-p53 pathway. (**A**) Schematic representation of the experimental strategy used for examining the effect of *Runx3* restoration in KPR*^L/F^* mice. The KPR*^L/F^* mice bear the *Rosa26R-Tomato* (Tomato*), *K-Ras^LoxP-Stop-LoxP-G12D/+^* (*K-Ras^LSL-G12D/+^*), *p53^flox/flox^*, *Runx3^flox/FSF^*, and *Flp^ERT2^* alleles. In the KPR*^L/F^* mice, *Ad-Cre* infection activated *K-Ras*, deactivated *p53* and *Runx3*, and induced lung ADs/ADCs. Treatment with tamoxifen (TAM) restored *Runx3* by activating *Flp^ERT2^*, which deleted the *Frt-Stop-Frt* cassette from the *Runx3^FSF^* allele. (**B**) Gross images of Tomato fluorescence emitted under UV light from the lungs of KPR*^L/F^* mice 2 weeks after *Ad-Cre* infection. (**C**) Microscopic images of lung tumors developed in KR*^L/F^* mice and KPR*^L/F^* mice 2 weeks after *Ad-Cre* infection. Lung tumors subjected to HE staining are shown. (**D**) Survival curves of KR*^L/F^* mice and KPR*^L/F^* mice infected with *Ad-Cre*. The median survival of the mice is shown on the right. *p* = *p*-value. (**E**) Survival curve of KPR*^L/F^* mice infected with *Ad-Cre*. Two weeks after *Ad-Cre* infection, the mice were fed normal food (KPR*^L/F^*-TAM(-), *n* = 5) or tamoxifen-containing food (KPR*^L/F^*-TAM(+), *n* = 5) for two weeks, followed by normal food in all mice. The median survival of the mice is shown on the right. (**F**) Schematic representation of the experimental strategy used for establishing KPR^-^ and KPR^restored^ cell lines from lung ADCs developed in KPR*^L/F^*-TAM(-) mice. (**G**) KPR^-^ cells were cultured in the presence or absence of 4-OHT and harvested at the indicated time points. Expression of Runx3 and formation of the R-point-associated activator (Rpa-RX3-AC) complex was measured using immunoprecipitation (IP) followed by immunoblotting (IB). Induction of Arf expression was measured through IB. (**H**) The KPR^-^ and KPR^restored^ cell lines were transfected with empty vector (Vec) or p53-expressing plasmids. The expression levels of Runx3, Arf, p53, and cleaved Caspase-3 were detected using IB.

**Figure 6 cells-12-02438-f006:**
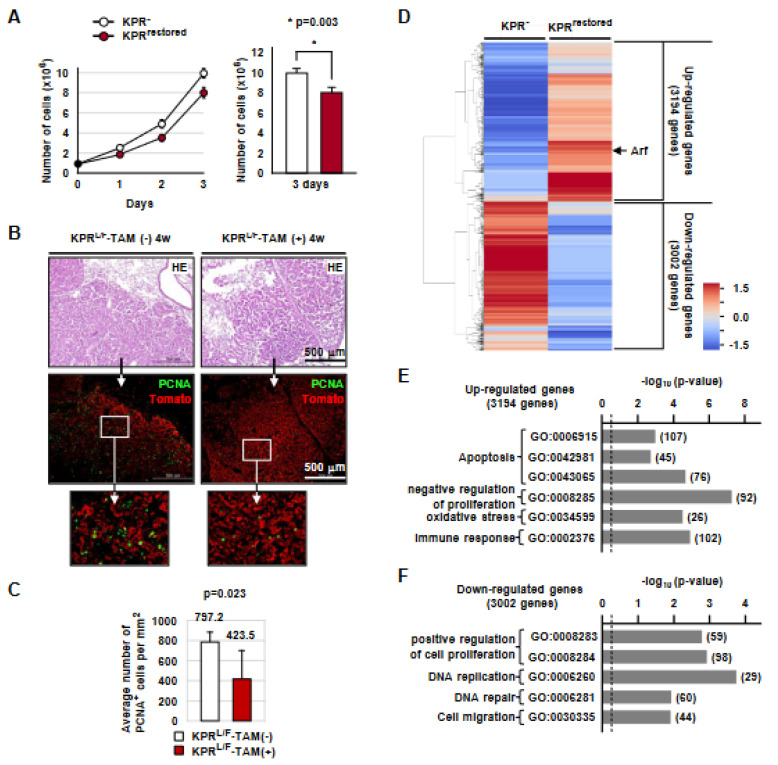
*Runx3* restoration inhibits the proliferation of lung ADC cells. (**A**) KPR^-^ and KPR^restored^ cells were cultured and the cell numbers were analyzed. The average numbers of KPR^-^ and KPR^restored^ cells 3 days after culture are shown on the right (*p* = *p*-value). (**B**) Microscopic images of KPR*^L/F^*-TAM(-)-4w and KPR*^L/F^*-TAM(+)-4w mouse lungs subjected to HE, PCNA, and Tomato staining. (**C**) The average numbers of PCNA-positive cells per mm² of cancer regions. (**D**) Heatmap showing genes up or downregulated by *Runx3* restoration. The fold change of each gene was converted to the log_2_ value to generate the heatmap. (**E**,**F**) The major signaling categories of upregulated and downregulated genes in *Runx3* restoration are shown. The numbers in the brackets indicate the number of genes.

## Data Availability

The RNA-seq data are available under accession number GSE243968 in the Gene Expression Omnibus (GEO).

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
