# Peer review of "Runx3 Restoration Regresses K-Ras-Activated Mouse Lung Cancers and Inhibits Recurrence"

_cells, 2023, doi:10.3390/cells12202438_

Round 1

Reviewer 1 Report

The manuscript by Lee YS, Lee JY, Lee JW et al entitled “Runx3 restoration regresses K-Ras–activated mouse lung cancers and inhibits recurrence” is focused on tumor suppressive function of Runx3 against K-Ras-activated lung carcinogenesis. In this study, conditional restoration of Runx3 in an established K-Ras-activated mouse lung cancer model suppressed both AD and ADC, reduced cancer recurrence, and significantly increased mouse survival, providing in vivo evidence supporting that RUNX3 is a therapeutic tool for the treatment of K-Ras-activated lung cancer. The data presented is compelling, with a good combination of mouse lines.

Concerns:

1. In Fig.2A, Arf and p53 in particular are unclear. These figures should be enlarged or made clearer.

2. The text describes Fig.2D, but Fig.2D is not in Figure 2. This is a major problem.

3. Runx3 is associated with phosphorylation at Ser15 of p53 to activate p53. Therefore, phosphorylation of p53 should also be evaluated by western blot.

4. In Fig.4B, the number of dots and the ‘n’ number do not seem to match. Especially put on the white dots. The others should also be checked.

5. Authors state that the survival of the model mice was indeed inversely proportional to the number of functional Runx3 alleles. Hence, functional Runx3 prolongs the survival of mouse models. It should be described as "directly" not "inversely." in line 393

6. Authors claim that Runx3 induces apoptosis in a p53-dependent manner and that the induction of apoptosis by Runx3 is attenuated in p53-deficient tumor cells. However, significant prolongation of mouse survival by Runx3 has also been shown in p53-deficient mice in Fig.5E. Is there a p53-independent mechanism of apoptosis by Runx3? Or more explanation of this p53-independent tumor suppressive function of Runx3 needs to be added.

7. Fig.5E should be Fig.5D in line 505.

Author Response

Comment

  1. In Fig.2A, Arf and p53 in particular are unclear. These figures should be enlarged or made clearer.

Answer

We appreciate for the Reviewer’s comment. We submitted figures with high resolution. However, we found that the microscopic images in the manuscript shown to reviewer are poor. It appears that the resolutions of the figures were reduced during the process by the publisher. Therefore, we showed enlarged microscopic images as Supplementary Figures (Suppl. Fig. 5).

Comment

  1. The text describes Fig.2D, but Fig.2D is not in Figure 2. This is a major problem.

Answer

We appreciate for the Reviewer’s comment. It was our mistake. We corrected the figure numbers (Fig. 2D to Fig. 1D, lines 314, 315, 456 and 465).

Comment

  1. Runx3 is associated with phosphorylation at Ser15 of p53 to activate p53. Therefore, phosphorylation of p53 should also be evaluated by western blot.

Answer

p53 is activated through two independent pathways: DNA damage and oncogene activation. As the reviewer commented, RUNX3 is closely involved in DNA damage-dependent phosphorylation of p53 at Ser-15 and acts. However, there is no report about the phosphorylation at Ser15 of p53 by oncogene activation. Our manuscript describes role of RUNX3 in oncogenic RAS activation. Therefore, we think evaluation of phosphorylation of p53 is not necessary for our paper.

Comment

  1. In Fig.4B, the number of dots and the ‘n’ number do not seem to match. Especially put on the white dots. The others should also be checked.

Answer

The number of dots does not indicate number of mice ‘n’. It indicates the number of counting of the live mice. Therefore, the figures for the mouse survival curve are correct.

Comment

  1. Authors state that the survival of the model mice was indeed inversely proportional to the number of functional Runx3 alleles. Hence, functional Runx3 prolongs the survival of mouse models. It should be described as "directly" not "inversely." in line 393

Answer

We appreciate for the Reviewer’s comment. We corrected our manuscript as the reviewer commented (“inversely” to “directly”, lines 405 and 409).

Comment

  1. Authors claim that Runx3 induces apoptosis in a p53-dependent manner and that the induction of apoptosis by Runx3 is attenuated in p53-deficient tumor cells. However, significant prolongation of mouse survival by Runx3 has also been shown in p53-deficient mice in Fig.5E. Is there a p53-independent mechanism of apoptosis by Runx3? Or more explanation of this p53-independent tumor suppressive function of Runx3 needs to be added.

Answer

We appreciate for the Reviewer’s comment. As the reviewer commented, we claimed that Runx3 induces apoptosis in a p53-dependent manner. However, Runx3 restoration could also prolong mouse survival in p53-deficient mice. These results suggested that the role of Runx3 is not limited in inducing apoptosis. To understand the additional role of RUNX3 we measured cell proliferation and found that Runx3 inhibits proliferation in a p53-independent manner. These results were described in Results (3.8. Runx3 restoration inhibits proliferation of K-Ras-activated lung tumor cells in a p53-independent manner, lines 522-560).

Comment

  1. Fig.5E should be Fig.5D in line 505.

Answer

We appreciate for the Reviewer’s comment. It was our mistake. We corrected the figure numbers (line 517).

Reviewer 2 Report

The authors established a novel Runx3 KO mouse, which allowed the restoration of Runx3 via Cre recombinase. In the context of K-Ras–activated lung cancer the authors demonstrated that Runx3 restoration increases the survival of the mice. Mechanistically they demonstrate via TUNEL staining that Runx3 restoration induces apoptosis in the in the lung cancer, likely via the Arf-p53 pathway. Further they showed that recurrent lung cancer samples often show a hyper-methylation of the runx3 promoter, suggesting that inactivation of runx3 is an important to maintain the K-Ras-activated lung cancer. This indicates a tumor-suppressive role of Runx3 in this cancer type. Further experiment using p53flox-mice suggesting this effect is mostly dependent on p53 but may also involve other pathways.

Together this study describes a novel role of Runx3 for K-Ras–activated lung cancer, which is overall convincing. The authors propose that Runx3 could be used as therapeutic tool. However, the manuscript lacks novel mechanistic insights and may therefore only be of limited interest for the scientific community. Also, it remains unclear in which way the authors envision how Runx3 can be exploited in cancer therapy.

Major points:

1) The RNA-Seq in Fig. 6D. appear to have been performed only in one replicate. It is common to perform RNA-Seq experiments with at least 3 biological replicates. The authors should comment about the limitations of this experiment in the manuscript.

2) The figures a generally very blurry. The quality of the figures should be improved for publication. This would be particularly essential for the microscopy figures, which are currently un-interpretable.

3) The authors should discuss more clearly in which way they believe Runx3 can be used as a therapeutic tool.

Some minor comments:

1) There should be a space before the header for 3.8.

2) In the material and methods it should be added how the gene ontology analysis of the RNA-Seq data has been performed.

Author Response

Comment

The RNA-Seq in Fig. 6D. appear to have been performed only in one replicate. It is common to perform RNA-Seq experiments with at least 3 biological replicates. The authors should comment about the limitations of this experiment in the manuscript.

Answer

We appreciate for the Reviewer’s comment. We agree to the reviewer’s comment. We indicated in the manuscript as follows "Further study is required to identify statistically meaningful target genes of the Runx3-induced p53-independent inhibition of proliferation” (lines 559-560).

Comment

2) The figures a generally very blurry. The quality of the figures should be improved for publication. This would be particularly essential for the microscopy figures, which are currently un-interpretable.

Answer

We appreciate for the Reviewer’s comment. We submitted figures with high resolution. However, we found that the microscopic images in the manuscript shown to reviewer are poor. It appears that the resolutions of the figures were reduced during the process by the publisher. Therefore, we showed enlarged microscopic images as Supplementary Figures (Suppl. Fig. 2, 4, 5 and 10).

Comment

3) The authors should discuss more clearly in which way they believe Runx3 can be used as a therapeutic tool.

Answer

We appreciate for the Reviewer’s comment. We described in the manuscript why Runx3 can be used as a therapeutic tool as follows “In this study, Runx3-restored mouse lung cancer did not recur until 10 weeks after regression (14 weeks after Runx3 restoration). The recurred tumor beared spontaneous inactivation of the restored Runx3 allele without secondary oncogene activation. These results demonstrate that expression of Runx3 could be helpful for the treatment of lung cancer and for achieving sustained regression.” (lines 604-608).

 Comment

4) There should be a space before the header for 3.8.

Answer

We appreciate for the Reviewer’s comment. We corrected as the reviewer commented (line 521).

 Comment

5) In the material and methods it should be added how the gene ontology analysis of the RNA-Seq data has been performed.

Answer

We appreciate for the Reviewer’s comment. As the reviewer commented we described how the gene ontology analysis of the RNA-Seq data has been performed as follows “Data mining and graphic visualization were performed using ExDEGA (Ebiogen Inc., Seoul, Korea). Gene clustering was analyzed using DAVID Bioinformatics Resources 2021 [32].” (lines 221-223).